

# Lightness induction enhancements and limitations at low frequency modulations across a variety of stimulus contexts

Louis Nicholas Vinke[1,2] and Arash Yazdanbakhsh[1,2,3]

[1] Graduate Program for Neuroscience, Boston University, Boston, MA, USA
[2] Center for Systems Neuroscience (CSN), Boston University, Boston, MA, USA
[3] Department of Psychological and Brain Sciences, Boston University, Boston, MA, USA

Corresponding author
Arash Yazdanbakhsh,
yazdan@bu.edu

## ABSTRACT

Lightness illusions are often studied under static viewing conditions with figures varying in geometric design, containing different types of perceptual grouping and figure-ground cues. A few studies have explored the perception of lightness induction while modulating lightness illusions continuously in time, where changes in perceived lightness are often linked to the temporal modulation frequency, up to around 2–4 Hz. These findings support the concept of a cut-off frequency for lightness induction. However, another critical change (enhancement) in the magnitude of perceived lightness during slower temporal modulation conditions has not been addressed in previous temporal modulation studies. Moreover, it remains unclear whether this critical change applies to a variety of lightness illusion stimuli, and the degree to which different stimulus configurations can demonstrate enhanced lightness induction in low modulation frequencies. Therefore, we measured lightness induction strength by having participants cancel out any perceived modulation in lightness detected over time within a central target region, while the surrounding context, which ultimately drives the lightness illusion, was viewed in a static state or modulated continuously in time over a low frequency range (0.25–2 Hz). In general, lightness induction decreased as temporal modulation frequency was increased, with the strongest perceived lightness induction occurring at lower modulation frequencies for visual illusions with strong grouping and figure-ground cues. When compared to static viewing conditions, we found that slow continuous surround modulation induces a strong and significant increase in perceived lightness for multiple types of lightness induction stimuli. Stimuli with perceptually ambiguous grouping and figure-ground cues showed weaker effects of slow modulation lightness enhancement. Our results demonstrate that, in addition to the existence of a cut-off frequency, an additional critical temporal modulation frequency of lightness induction exists (0.25–0.5 Hz), which instead maximally enhances lightness induction and seems to be contingent upon the prevalence of figure-ground and grouping organization.

## INTRODUCTION

Visual illusions which induce illusory lightness percepts of a central area with uniform reflectance are highly dependent upon the static luminance profile of adjacent components in the stimulus (*Kingdom, 1997*). These types of illusions have been mostly studied under static viewing conditions (*Blakeslee & McCourt, 1999*; *Heinemann, 1955*; *White, 1979*; *Williams, McCoy & Purves, 1998*), wherein the perceived lightness is compared across various static contexts. Few studies have explored the perception of lightness induction illusions while modulating lightness illusions continuously in time (*De Valois et al., 1986*; *Penacchio, Otazu & Dempere-Marco, 2013*; *Rossi & Paradiso, 1996*, *1999*). When certain components of these illusions are modulated in time, human observers often report induced lightness modulation occurring within other components containing a static uniform luminance and reflectance, which is linked to the current modulation frequency. The temporal modulation of lightness induction has been shown to be a relatively slow process, operating up until 2–4 Hz (*De Valois et al., 1986*; *Rossi & Paradiso, 1996*), after which a flickering percept may persist, followed by a complete loss of lightness induction as the modulation frequency is further increased. However, alternative experimental paradigms support the existence of a much faster process operating up to at least 24 Hz (*Blakeslee & McCourt, 2008*).

An under-appreciated result observed in these few continuous lightness modulation studies is a relatively large increase in lightness induction when simply comparing static and slow continuous modulation viewing conditions (*Rossi & Paradiso, 1996*, *1999*), which in one instance did not actually bear out in the results, despite the authors anecdotal observations (*De Valois et al., 1986*). While much attention has been focused on cut-off frequency states at higher temporal modulation frequencies, currently no studies have explicitly explored the potential existence for another critical lightness induction state change at very low temporal modulation regimes (<0.5 Hz). The possibility of a significant change in lightness induction, by simply introducing a relatively slow continuous modulation, remains poorly understood and requires a thorough investigation.

Various types of visual illusions in a static context have been studied using a perceptual grouping framework, which is largely based upon Gestalt grouping principles (*Wagemans et al., 2012*). For lightness illusions, grouping principles, such as generalized common fate, element connectedness, and uniform connectedness, can be considered to be the primary driving forces behind many perceptual illusions (*Gilchrist, 2014*). With the introduction of temporal modulation, common temporal structure and an alternate interpretation of the generalized common fate principle have also been considered (*Guttman, Gilroy & Blake, 2007*; *Lee & Blake, 1999*; *Sekuler & Bennett, 2001*). Lightness illusions have also been differentiated based on the figure-ground cues present at border junctions, which are thought to provide local occlusion cues to the observer (*Anderson, 1997*, *2003*). In this case, different junction types are thought to facilitate the perception of depth relationships between different objects in the scene. Common junction types include L-junctions (Simultaneous Contrast illusion), T-junctions (Münker–White illusion) and X-junctions (Checkerboard illusion). The L- and T-junctions are thought to

provide strong cues for figure-ground segregation (*Ross & Pessoa, 2000*; *Todorović, 1997*) by highlighting a shared border between two or more separate regions, that is ultimately perceived as an occluding edge of the group of regions sharing that border. As a result of border-ownership assignment, some regions are then more likely to be perceived as the background, with the occluding region being the figure. This can occur despite retinal adjacency of stimulus components under 2D viewing conditions (*Layton, Mingolla & Yazdanbakhsh, 2012*; *Zaidi, Spehar & Shy, 1997*). Conversely, X-junctions tend to promote ambiguity between the perceptual inferences of occlusion (multiple planes), adjacency (single plane), and/or transparency (*Adelson, 1993*). While the dynamics that may be present in a visual illusion can be interpreted using certain perceptual grouping principles, questions still remain regarding how different types of figure-ground cues are affected by temporal dynamics, and the extent to which perceptual grouping principles may reinforce these cues to accentuate lightness induction under continuous temporal modulation viewing conditions relative to static viewing.

There are two main goals in the present study: (1) to explicitly examine whether lightness induction undergoes any critical change between static viewing and continuous modulation states; (2) to explore the extent to which different perceptual grouping principles and figure-ground cues may contribute to any critical changes we observe across modulation conditions. While the visual illusions examined in this study are differentiated based on figure-ground junction type, the results are interpreted in terms of both perceptual grouping and figure-ground definitions. In general, improving our understanding of how temporal modulation interacts with grouping and segmentation cues applicable to a particular visual lightness illusion can offer further insight into the early visuocortical processes involved in the percept of lightness illusions in humans.

## MATERIALS AND METHODS

### Participants

Nine participants (three female, ages 18–45) with normal or corrected-to-normal vision were enrolled in this study. Participants were recruited from the Boston University community. All participants received class credit or monetary compensation for their participation. This study was approved by the Boston University Charles River Campus Institutional Review Board (3651E). All participants were fully informed about the goals of the study and provided written consent before their participation.

### Stimuli

All stimuli were generated in MATLAB 2013b (*TheMathWorks Inc., 2012*), using the Psychophysics Toolbox (*Brainard, 1997*). A LCD monitor (Hanns-G Hi221; mean luminance: 50.19 cd/m$^2$; frame rate: 60 Hz) was used to display the stimuli, which was linearized using gamma-correction generated from photometer measurements (LS-100; Konica Minolta, Chiyoda City, Tokyo, Japan). The entire display region subtended 27.6° × 42.78° (height × width) of the participant's visual angle. All stimulus types are depicted in Figs. 1A–1D. The gray target region was located at central fixation, and was 2.3° × 2.3° (height × width) for all stimulus types. The dimensions for the Simultaneous

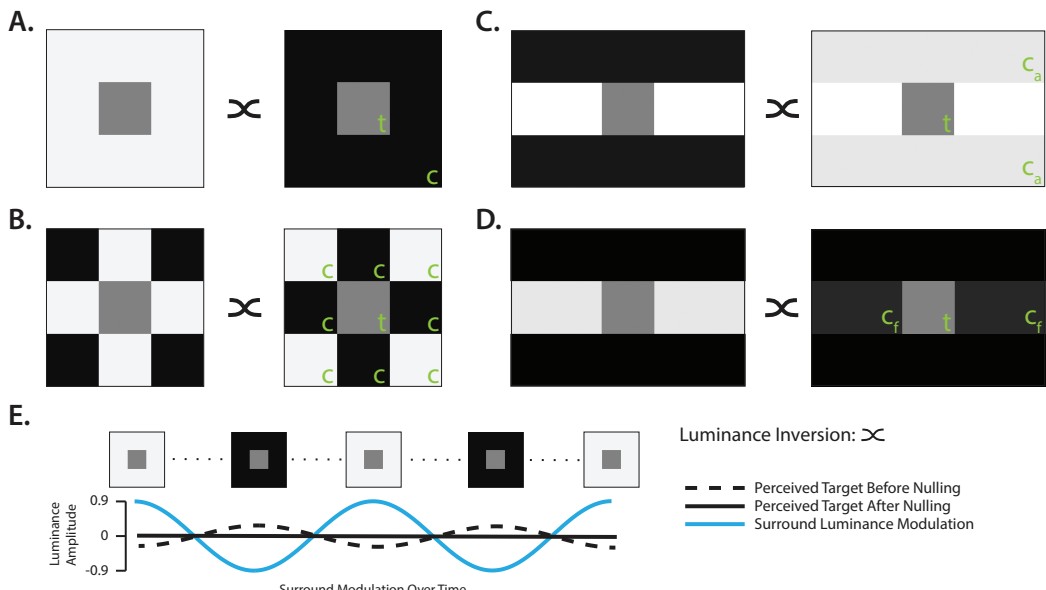

**Figure 1 Experimental stimuli and procedure.** Examples of lightness induction stimuli presented to participants, and nulling task procedure. For all experimental stimuli, the stimulus context (c) was temporally modulated while participants adjusted the central target (t) region to eliminate any perceived lightness modulation. Two possible extremes are depicted for each stimulus, each illustrating a pair of stimuli with the surround inverted. (A) Simultaneous Contrast stimulus. (B) Checkerboard stimulus. (C) Münker–White assimilator stimulus. Only the assimilator ($c_a$) portions of the stimulus context were varied on temporal modulation trials. Modulated components are depicted at a decreased contrast for viewing purposes. (D) Münker–White flanker stimulus. Only the flanker ($c_f$) portions of the stimulus context were varied on temporal modulation trials. Modulated components are depicted at a decreased contrast for viewing purposes. (E) Example time course of the simultaneous contrast stimulus during the nulling procedure. At trial onset, the central target region had a fixed mean luminance. However, participants experienced an induced lightness modulation within the target region, dependent on the temporal modulation frequency (dashed line). By adjusting the counter-phase luminance offsets of the target region, participants could reduce and eliminate any perceived lightness modulation induced by stimulus context modulation, and effectively made the target region appear as having a fixed mean luminance over time (solid line). The maximum and minimum contrast for all four stimulus conditions were limited to 90% and −90% respectively.

Contrast (SC) and Checkerboard (Chk) stimulus types was 6.9° × 6.9° (height × width). The Münker–White stimulus type encompassed an assimilator ($MW_a$) and a flanker ($MW_f$) subtype, each measuring 6.9° × 10.4° (height × width). The dimensions of the Münker–White stimuli were chosen such that the assimilator area matched the total area of the other stimulus types. As a result, the flanker area was approximately one third the size of the assimilator area in order to maintain the general Münker–White configuration and the square aspect ratio of the gray target region.

## Experimental design and procedure

Previous lightness induction studies have employed paradigms where participants are asked to match the magnitude of the lightness illusion to a probe (*De Valois et al., 1986*; *Zaidi, Spehar & Shy, 1997*). This often proves to be a difficult task for participants, as task difficulty tends to scale with increasing temporal frequency. To overcome the task difficulty imposed by matching paradigms, in the present study participants were asked to

perform a nulling paradigm task (*Krauskopf, Zaidi & Mandlert, 1986*; *Zaidi, Yoshimi & Flannigan, 1991*; *Adelson, 1993*) when viewing each stimulus type under a variety of temporal modulation conditions. Specifically, participants were asked to cancel out, as best as possible, any perceived modulation in lightness detected over time within the central target region located at fixation. While carrying out the nulling task, the context provided by the surrounding lightness illusion stimulus either alternated irregularly between two static states or was modulated continuously in time. Throughout any given trial, participants adjusted the perceived "grayness" of the central target region until they had eliminated, as best as possible, any perceived differences in lightness over time (see Fig. 1E). Additionally, participants were instructed to maintain central fixation throughout the entirety of each trial (red fixation annulus: 0.15° diameter). At the beginning of each trial, the central target region was randomly initialized at either the mean luminance, matching the background, or at ±25% relative to the mean luminance. In-between each trial, participants were presented with a 20 Hz flickering random white noise whole-field patch for 500 ms to eliminate any persisting afterimage related to stimuli viewed during the previous trial. Participants were given the opportunity to take a break after each block of trials.

On continuous surround modulation (CSM) trials, the surrounding context oscillated sinusoidally at one of several different logarithmically-spaced frequencies (0.25, 0.5, 1 and 2 Hz). The maximum and minimum luminance values of the dynamic stimulus segments during CSM trials ranged between 90% and 10% of the maximum luminance, respectively, in order to maintain physical borders between neighboring stimulus segments. For instance, this ensured that the $MW_a$ and $MW_f$ stimuli could not be mistaken for the SC stimulus at these extreme states (Figs. 1C and 1D). Each block of CSM nulling trials consisted of four stimulus types (SC, Chk, $MW_a$ and $MW_f$) × 4 modulation frequencies (0.25, 0.5, 1 and 2 Hz) repeated four times in a random order. Of the total nine participants, six completed 1 block of CSM trials (64 trials in total), while the first initial three participants completed 2 blocks of CSM trials (128 trials in total). An adequate number of frames per continuous modulation cycle was achieved for all stimulus types (0.25 Hz: 240 frames per cycle; 2 Hz: 30 frames per cycle), resulting in no aliasing or perceived jitter in stimulus modulation, ultimately creating a smooth appearance of sinusoidal modulation.

On static trials, participants had to manually switch between the stimulus context extremes, which was subject to a forced 2 s delay in order to prevent participants from switching at a frequency exceeding 0.25 Hz. In practice, all participants switched at a much slower and irregular rate. Each block of static nulling trials consisted of three stimulus types (SC, Chk, MW) repeated four times in a random order. Of the total nine participants, six completed 1 block of static trials (12 trials in total), while the first initial three participants completed two blocks of static trials (24 trials in total).

Since the static versions of the $MW_a$ and $MW_f$ stimuli were identical, only one set of static nulling estimates were obtained for both of these stimulus types. This set of static nulling estimates was used for comparison against both $MW_a$ and $MW_f$ nulling estimates obtained separately across all the CSM conditions.

## Data analysis

The average nulling amplitude was calculated for all four stimulus types across all five modulation conditions (static and continuous). Nulling amplitude, on any given trial, was obtained as the final gray offset value of the central target region from mean luminance, and calculated in percent difference relative to the mean display luminance. The nulling amplitude was analogous to the percent difference from mean luminance, where a nulling amplitude of 0% corresponded to a gray value of the central target region which was no different from the mean display luminance. An intermediate nulling amplitude, for example, 20%, corresponded to a gray offset value of the central target region which was maximally 20% lighter and darker than the mean display luminance during modulation (static or continuous). Two-way between-subjects ANOVA statistical tests were performed to determine whether any significant main effects and interactions of stimulus type and temporal modulation on group-wise mean nulling amplitude measurements existed. Furthermore, to determine whether any significant trends existed across log-spaced modulation frequency conditions, a linear function was fit to the data, at the subject-wise level, using a least-squares fitting procedure. The Bonferroni method was used to correct for multiple comparisons (alpha = 0.0125) for all subsequent $t$-test statistical analyses.

## RESULTS

In order to investigate the combined effects of continuous surround modulation (CSM) frequency and segmentation cues on lightness induction, perceived lightness modulation induced in a central target region was measured across multiple modulation frequencies using a nulling paradigm. The results of a two-way between-subjects ANOVA (excluding the static condition) demonstrated a main effect of CSM frequency ($F_{3,\ 128} = 7.73$, $p < 0.001$), and a main effect of stimulus type ($F_{3,\ 128} = 11.19$, $p < 0.001$). No significant interaction between CSM frequency and stimulus type was found ($F_{9,\ 128} = 0.42$, $p = 0.923$). In general, nulling amplitude decreased as CSM frequency increased, indicating that the strongest perceived lightness induction occurred at lower CSM frequencies (Fig. 2A). For the slowest CSM frequency condition, the L-type junction stimulus (SC) had the highest mean nulling amplitude (22.6% ±6.1), indicating this stimulus type produced the strongest lightness induction perception. Stimuli containing T-type junctions also produced relatively high mean nulling amplitudes ($MW_f$: 19.8% ±9.0; $MW_a$: 18.1% ±11.9). The X-type junction stimulus (Chk) had the lowest mean nulling amplitude (10.1% ±6.5), producing the weakest lightness induction, approximately half as strong as the L-type junction stimulus (SC).

## Lightness induction trends across continuous surround modulation frequencies

To further examine the effect of CSM frequency on lightness induction, a linear function was fit to the mean group nulling amplitude measurements for each stimulus condition across all CSM frequencies converted to log-space. The slope parameter of the fitted linear function reflects any linear trend in lightness induction across the log-spaced CSM

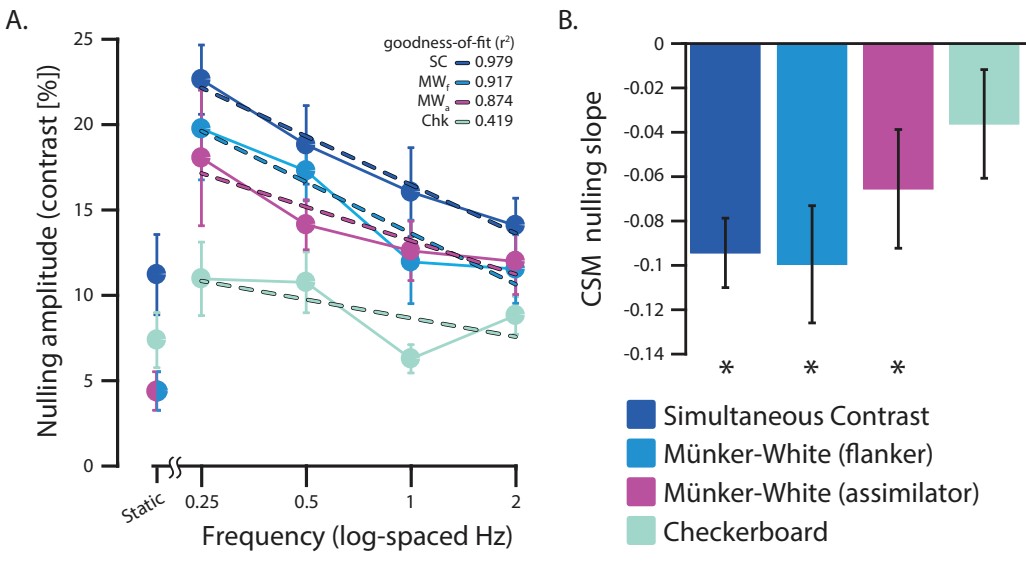

**Figure 2 Effects of surround modulation and stimulus type on lightness induction.** Group-wise ($n = 9$) effects and trends of temporal modulation frequency on nulling amplitudes across all stimulus conditions. (A) Mean group nulling amplitude across the static and continuous surround modulation frequency conditions for all stimulus types. A nulling amplitude of 0% would indicate no perceived lightness induction for that specific experimental condition and stimulus type. (B) Mean group slope estimates representing any trends in nulling amplitude measurements across the continuous surround modulation (CSM) frequency conditions. All error bars represent one standard error of the mean (asterisks denote $p < 0.01$).

frequencies (Fig. 2B). Group-averaged slope estimates were significantly different from zero for the L-type junction stimulus (SC: ($t(8) = -6.0297$, $p < 0.001$)) and for both flanker (MW$_f$: ($t(8) = -3.7663$, $p = 0.0055$)) and assimilator (MW$_a$: ($t(8) = -2.4478$, $p = 0.0401$)) T-type junction stimuli, while no significant difference was found for the X-type junction stimulus (Chk: ($t(8) = -1.4791$, $p = 0.1774$)). In general, all significant group-averaged slope estimates were negative, reflecting a decrease in perceived lightness induction as the CSM frequency was increased. These results indicate that the degree of perceived lightness induction produced by L-type (SC) and both T-type (MW$_f$ and MW$_a$) stimulus configurations is particularly contingent upon CSM frequency.

## Effect of continuous surround modulation vs. static viewing on lightness induction

The degree to which even a slight degree of continuous surround temporal modulation can alter lightness induction was examined by comparing the nulling amplitude responses between the static condition and the slowest continuous surround modulation (CSM) condition (0.25 Hz, 1 cycle per 4 s), for all stimulus types (Fig. 3). Significant differences between perceived lightness in the static and (slowest) CSM conditions were found for the L-type junction stimulus (SC: ($t(8) = 4.4485$, $p = 0.0021$)), and both T-type junction stimuli (MW$_f$: ($t(8) = 6.0532$, $p < 0.001$); MW$_a$: ($t(8) = 3.5697$, $p = 0.0073$)). No significant difference was found for the X-type junction stimulus (Chk: ($t(8) = 1.5204$, $p = 0.16$)). A comparison of effect sizes (Cohen's $d$ reported) indicates that the flanker T-type junction

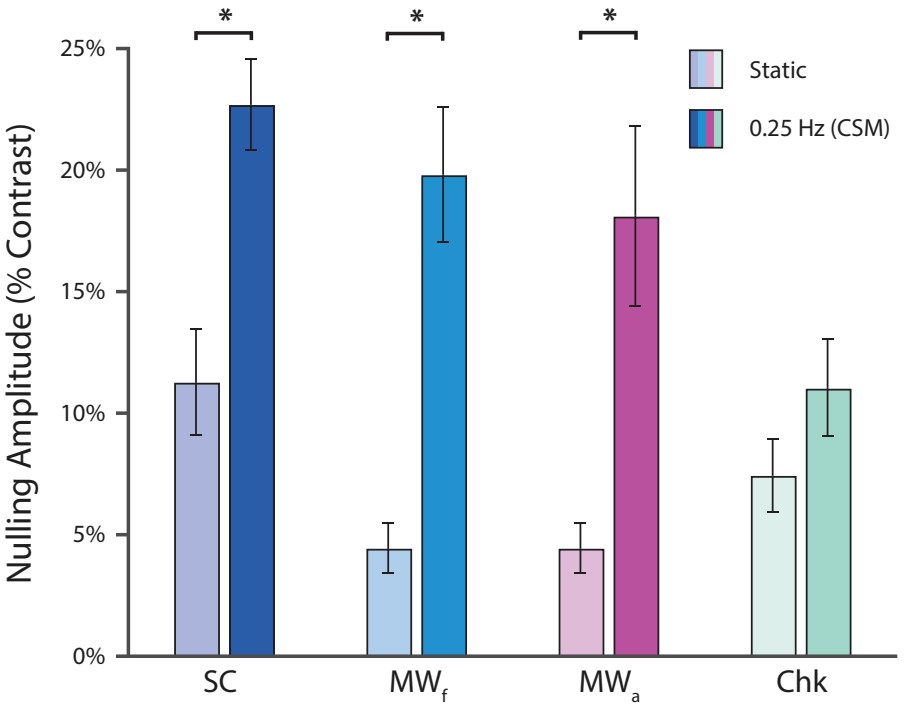

**Figure 3 Effect of discrete vs. continuous modulation on lightness induction.** Group-wise effects of the static and continuous surround modulation conditions on lightness induction across all stimulus conditions. Mean group nulling amplitude for the static condition and the slowest continuous surround modulation (CSM) condition (0.25 Hz) across all four stimulus types are depicted. An absolute nulling amplitude of 0% would indicate that no lightness induction was being perceived, and the central target was physically identical to the mean luminance background. Note that all static stimulus conditions have non-zero mean nulling amplitudes, confirming previous observations that lightness induction does occur with these particular surround organizations. The mean nulling amplitudes for both static Münker–White flanker and assimilator stimuli are identical because they were measured and computed from the identical trials (see "Methods" for details). All error bars represent one standard error of the mean (asterisks denote $p < 0.01$).

stimulus had the greatest nulling amplitude difference between static and the slowest dynamic modulation viewing conditions ($MW_f$: $d = 2.2706$), followed by the L-type junction stimulus (SC: $d = 1.7336$), assimilator T-type junction stimulus ($MW_a$: $d = 1.5619$), and lastly the X-type junction stimulus (Chk: $d = 0.6291$). These results strongly suggest that the perceived lightness induction of stimuli containing figure-ground cues based only on L-type and T-type junctions are augmented by slow (0.25 Hz) continuous surround temporal modulation.

## DISCUSSION

The findings and observations reported in previous studies examining the effects of temporal modulation on the cut-off frequencies of perceived lightness indicate significant changes in perceived lightness between static and very slow continuous modulation states when presenting stimuli analogous to the Simultaneous Contrast illusion (see (*De Valois et al., 1986*): "Discussion" and Fig. 4; (*Rossi & Paradiso, 1996*): Fig. 4; (*Rossi & Paradiso, 1999*): Fig. 9), indicating another critical change is present. However, since this

critical change was never the primary focus of these studies, a more thorough examination did not follow, despite the ubiquity of this finding in past temporal modulation studies. In this study, we used a nulling paradigm to measure the lightness illusion strength when contextual surrounds, containing different types of segmentation cues, were modulated continuously at multiple low frequencies (continuous modulation), or compared against a no temporal modulation condition (static), to investigate factors driving the significant changes in perceived lightness when little to no temporal modulation is present. We found significant increases in the perceived lightness during continuous low-frequency surround modulation compared to the static condition for illusions with contexts containing L-type junctions (SC), and T-type junctions ($MW_f$ and $MW_a$), with the exception of an illusion with X-type junctions (Chk). When manipulating the frequency of the continuous surround modulation, lightness illusion strength was found to be significantly higher compared to the static condition, before decreasing across frequency conditions (from low to high) for SC, $MW_f$ and $MW_a$ stimuli, while the Chk stimulus had no significant decreasing trends. The results also indicate that there exists a critical frequency greater than 0 Hz and less than or equal to 0.25 Hz (i.e., 1 cycle per 4 s) where the strength of the lightness illusions is strongest for SC, $MW_f$, and $MW_a$ stimuli, except in the case of the Chk stimulus. In general, this collection of findings suggests that dynamic changes in perceived lightness are dependent upon, at least in part, how apparent the figure-ground separation is perceived by the observer.

## Critical continuous surround modulation frequency

In this study we have identified a critical frequency located at a much lower continuous modulation frequency that differs from the cut-off frequency previously reported in the literature. As the temporal modulation frequency decreases and approaches a static state, the lightness induction strength increases. Conceivably, this effect persists until the temporal modulation frequency is no longer discriminable from the static state, which could be considered to be the Just Noticeable Temporal Modulation (JNTM) threshold. It remains to be seen whether the maximal lightness induction strength occurs at the JNTM threshold. Alternatively, lightness induction may conform to an inverted-U shape or reach a plateau between 0.25 Hz and the JNTM (Fig. 4). Furthermore, upon confirming the existence of a JNTM threshold, it may be possible to identify a global maximum, or the most optimal frequency for producing the strongest modulation of lightness induction while keeping other factors fixed. In general, at a temporal modulation of 0.25 Hz we observed the largest lightness induction modulation, greater than that of the static condition for all stimuli, with the exception of the Chk stimulus. Previous studies investigating interactions between temporal modulation and lightness induction have tested surround contrast modulations with Michelson Contrasts up to 60% (*De Valois et al., 1986*) and 28.5% (*Rossi & Paradiso, 1996*). However, these previous studies did not devote much discussion to the substantial subjective differences between static and continuous temporal modulation conditions. The larger differences which we observed may in part be driven by modulating stimuli containing much greater contrast levels (maximum Michelson Contrast: 90%).
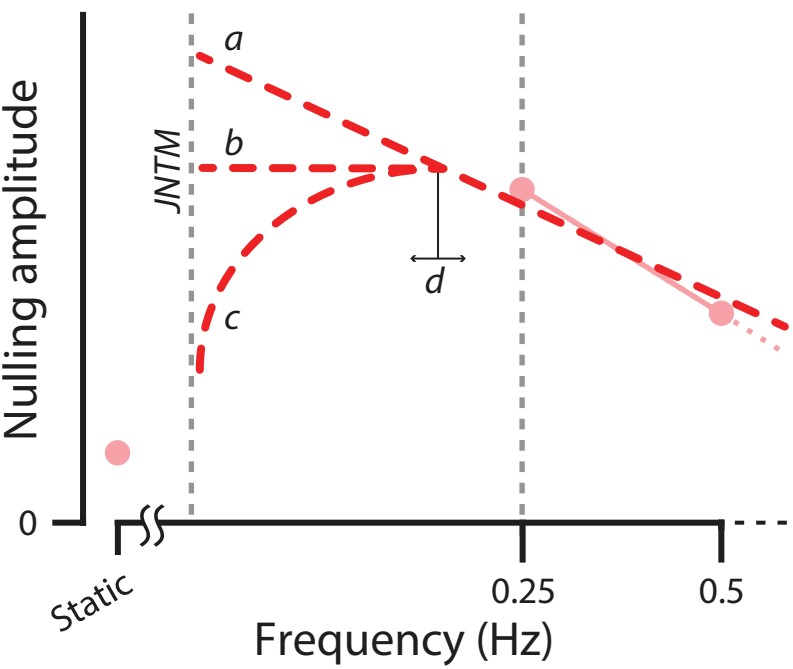

**Figure 4  Just noticeable temporal modulation (JNTM) limit.** Illustration depicting lightness induction strength trends as stimulus modulation frequency approaches the limit at which continuous modulation of the stimulus can no longer be reliably detected by the observer, and is perceived to be static in nature, termed the Just Noticeable Temporal Modulation (JNTM) limit. Over increasingly slower temporal modulation frequencies, the lightness induction strength is hypothesized to change in three qualitative ways: induction strength will (a) continue to increase, (b) reach a plateau, or (c) monotonically decrease. An inflection point (d) for the latter two hypothetical paths could occur at any point between the lowest temporal modulation condition in this study (0.25 Hz) and the JNTM limit. Data points from the Simultaneous Contrast condition are shown in low-opacity for illustrative purposes only.

Our results also demonstrate a decline in lightness induction strength as continuous surround modulation frequency is increased, consistent with the presence of a cut-off frequency above 2 Hz. All lightness illusions we examined did show a significant (SC, MW$_f$ and MW$_a$) or trending decline (Chk) in lightness induction strength as frequency increased. The effect of continuous surround modulation does seem to impact the SC and MW stimuli in a qualitatively different way. The surround configuration of these stimuli may provide a stronger and unambiguous figure-ground signal compared to the other stimuli, making the lightness induction more apparent despite all stimuli sharing certain general geometric similarities. Conversely, the lightness induction magnitude of the Chk stimulus seems to benefit the least from continuous surround modulation, suggesting that this type of modulation does little to establish or improve the already weak figure-ground visual cues present during the static viewing condition. Our results suggest that these perceived lightness differences may be dependent upon the integration time of the stimulus components at the cortical level. The variability in the effect of continuous surround modulation on lightness induction across stimulus types may be related to the number of additional levels of cortical processing necessary beyond the simple filtering properties of the retina and subcortical visual areas (*Jehee, Lamme & Roelfsema, 2007*).

## Extent of cortical recruitment for lightness induction

Evidence indicating that lightness induction can operate over distances larger than conventional retinal and geniculate receptive fields (*De Valois & Pease, 1971*; *Yund & Armington, 1975*) strongly suggests that lightness induction requires cortical recruitment in order for the lightness induction percept to emerge. More direct support for this claim comes from extracellular recordings, wherein striate cells were most likely to display activity phase-locked to flanker modulation of the lightness induction stimulus, despite the flanker being positioned outside of the conventional receptive field area (*Rossi & Paradiso, 1996*, *1999*). Interestingly, when the central gray region undergoing lightness induction is instead presented as uniformly black, then the phase-synchrony between striate activity and flanker modulation is extinguished (*Rossi & Paradiso, 1999*). This finding in particular suggests that lightness induction either relies upon modulation from lateral connections within visuocortical areas (*Macknik & Martinez-Conde, 2004*), or relies upon modulation from inter-areal connections across the visual hierarchy within the cortex (*Angelucci et al., 2002*; *Bullier, 2001*). In the latter case, the entire induction stimulus (center and flanker components) can both be contained within the larger conventional receptive field area in higher-order visual areas, and can be integrated as a whole before being propagated back to earlier visuocortical areas where the lightness induction phase-synchrony signature has been observed across all laminar layers (*Rossi & Paradiso, 1999*). Differences in edge-related single-unit activity coinciding with physical and illusory stimulus contours have been reported across early visuocortical areas in non-human primates (*Von der Heydt, Peterhans & Baumgartner, 1984*; *Zhou, Friedman & Von der Heydt, 2000*). However, a recent human fMRI study did not find distinct responses of neural populations representing the perceptual lightness modulation of uniform stimulus regions, while still finding strong support for edge-related responses (*Cornelissen et al., 2006*).

The hierarchical recruitment of cortical areas during lightness perception has also inspired modeling efforts which contain multiple banks of spatial filters at different orientations and spatial scales (spatial frequency) (*Blakeslee & McCourt, 1999*, *2004*; *Dakin & Bex, 2001*, *2003*). These models have been shown to make quantitative predictions for multiple visual illusions (Simultaneous Contrast, Münker–White, and others), which coincide with perceptual judgments, although it has recently been shown that this class of models do not capture the effect of narrowband noise on lightness induction (*Betz et al., 2015*). Interestingly, Blakeslee & McCourt, who have proposed and developed one of these prominent models (ODOG model), have also provided evidence for temporal modulation impacting brightness induction (grating induction stimulus), particularly at slow modulation frequencies. One of the primary motivations behind these models is to support the notion of a fast brightness induction process when perceiving these types of visual illusions, as opposed to relatively slower filling-in mechanisms. However, the results of our study, and the complimentary results of others, prompts the question of why more time for stimulus integration afforded during slower temporal modulation leads to stronger illusion effects. While the addition of a temporal component to the ODOG

model has been suggested in order to capture these temporal modulation effects (*Blakeslee & McCourt, 2011*), recent implementations of the model have yet to include temporal elements to capture dynamic changes in brightness induction (*Blakeslee, Cope & McCourt, 2016*).

## Lightness illusion differences

One previous study (*Robinson, Hammon & De Sa, 2007*) reported the mean perceived lightness differences in luminance (cd/m$^2$) from various lightness illusion studies which used matching paradigms, with Simultaneous Contrast having the largest effect (11.35 cd/m$^2$), followed by Checkerboard (5.67 cd/m$^2$), and Münker–White (4.18 cd/m$^2$). Our results produced a slightly different ranking, with SC having the largest effect (static: 5.62 cd/m$^2$; 0.25 Hz: 11.34 cd/m$^2$), followed by MW$_f$ (static: 2.21 cd/m$^2$; 0.25 Hz: 10.03 cd/m$^2$), MW$_a$ (static: 2.21 cd/m$^2$; 0.25 Hz: 9.08 cd/m$^2$), and Chk (static: 3.71 cd/m$^2$; 0.25 Hz: 5.52 cd/m$^2$). It is important to consider that absolute differences in lightness induction strengths between this study and others could be accounted for by differences in stimulus spatial frequencies and stimulus size. Regarding stimulus size, the modulation area of each stimulus type was not always equated. When all stimulus types are ranked by modulator area, the ranking roughly tracks the strength of the illusions under static viewing conditions (see Fig. 2A, static condition). However, when temporal modulation is introduced, the correspondence is lost. Based on modulation area alone, the MW$_f$ stimulus would be expected to show the smallest temporal modulation effect, but this is not what we observed. The MW$_f$ nulling amplitude difference between the static and the lowest frequency CSM condition was larger when compared to both the MW$_a$ and SC stimuli. Considering just the MW stimuli, despite the smaller area of the flanker modulator compared to the assimilator, the flanker modulator still generated a surprisingly strong lightness induction effect, which suggests that if one were to control for modulator area, then the flanker effect may have been even stronger. Relatedly, it has been reported that the lightness induction of the MW stimulus does not depend on the aspect ratio of the target region, and hence the overall aspect ratio of the stimulus (*Blakeslee & McCourt, 2004*; *Blakeslee, Padmanabhan & McCourt, 2016*; *Güçlü & Farell, 2005*), but conflicting reports do exist (*Mitra et al., 2018*). Although, the magnitude of the MW stimulus has been shown to be resilient across a wide range of lower spatial frequencies, and demonstrates an increase across higher spatial frequencies (*Blakeslee & McCourt, 2004*; *Helson & Rohles, 1959*; *White, 1979*). It is also worth-noting that the aspect ratio of the SC illusion has been shown to systematically influence lightness induction strength (*Shi et al., 2013*; *Yund & Armington, 1975*).

How might various perceptual grouping principles and figure-ground cues contribute to the differences in lightness induction we observe across experimental conditions? While the perceptual grouping principles mainly provide a guide for differentiating figural regions from background regions, the grouping organization they promote identifies potential candidates for figure or background designations. Some of the lightness illusions can adhere to the generalized common fate principle (similarity in luminance) (*Sekuler & Bennett, 2001*), and the uniform connectedness principle (*Palmer & Rock, 1994*).

Considering the similarity principle, elements with similar or identical luminance, will facilitate grouping for all three illusion types in this study, with the mutually exclusive regions of each illusion having been initially identified based on local uniform luminance properties (uniform connectedness). Unlike the MW illusion, the Chk illusion does not benefit from the element connectedness principle, since none of the distinct uniform regions which are grouped based on luminance similarities, share any continuous borders with one another. The lack of evidence for shared borders within each luminance group of the Chk illusion is also conveyed with the figure-ground ambiguity associated with X-junctions. The flanker and assimilator MW illusion types, differ in regards to grouping by element connectedness. The flanker regions clearly share common (horizontal) occluding borders, the ownership of which is assigned to the assimilator regions given the presence of the T-junctions, defining the assimilator regions as figural components, and the flanker regions as background or as part of a different plane. Due to the lack of multiple regions in the SC illusion (excluding the target region), the element connectedness principle does not apply. Under temporal modulation conditions, a perceptual grouping principle based on temporal structure comes online (*Guttman, Gilroy & Blake, 2007*; *Lee & Blake, 1999*). Simultaneous luminance changes in distinct regions occurring over time further promotes the existing grouping organization already present under static viewing conditions. Presumably, the temporal modulation of luminance does not have a direct impact on the figure-ground organization supported by the different junction types, since the stimulus component borders remain intact. However, our results demonstrate that temporal modulation further reinforces the segregation of the figural and background groupings.

## Nulling vs. matching tasks

In contrast to previous studies using matching paradigms to examine temporal effects on lightness illusions (*De Valois et al., 1986*; *Rossi & Paradiso, 1996*; *Zaidi, Spehar & Shy, 1997*), we used a nulling paradigm which offers multiple advantages: (1) No eccentricity confounds, (2) no short-term memory confounds, and (3) no covert attention demands. Since participants can perform the nulling procedure while maintaining fixation throughout the task/trial, the perceived lightness measurements are being made at fixation relative to the surrounding contexts, and generating no confounding issues of eccentricity or short-term memory. Matching paradigms require participants to saccade back and forth between a target and matching stimulus while remembering the 'grayness' in order to make their judgments, or alternatively, participants covertly attend to a comparison target at a distal spatial location, often located several degrees out from fixation. In this particular nulling paradigm, any potential confounds related to the proximity of CSM frequencies to flicker fusion thresholds, wherein a rapidly changing stimulus is perceived as having a steady appearance, are avoided since the highest CSM frequency measured (2 Hz) is roughly two orders of magnitude lower than the flicker fusion threshold (65–100 Hz) (*Davis, Hsieh & Lee, 2015*; *Roberts & Wilkins, 2013*). Furthermore, while the mean background luminance and spatial arrangement of each stimulus type was held constant across temporal modulation and static viewing conditions, the degree to which

adaptation may have been perturbed within each viewing condition may not be completely analogous due to the different modulation changes being encountered. During static viewing, participants flipped between stimulus extremes fairly regularly, at most every 2 s (imposed limit) up to every 3–4 s. Throughout each trial participants were also systematically altering the target patch luminance between each stimulus flip. These aperiodic stimulus modulations may have also perturbed the ability of adaptation to any specific target gray level to take place, similar to modulation changes encountered during the CSM conditions. However, it has been previously reported that following sustained adaptation to a particular luminance level, the presentation time of probe stimuli over 1–4 s had minimal influence on apparent brightness judgments (*Saunders, 1968*).

Certain previous studies employing matching paradigms (*De Valois et al., 1986*) have noted that some participants expressed difficulty with the matching task due to an asymmetry in the lightness induction strength, specifically with a larger perceived induction strength as the modulation approached the darkest state. Participants who experienced this asymmetry had difficulty identifying a single value which matched the target. One participant in our experiments reported a similar experience, indicating that this phenomenon can also apply to the nulling paradigm. The prevalence of this phenomenon suggests that despite the fact that the temporal modulation of the stimulus surround is centered at the mean luminance, which evenly bisects the maximum and minimum surround luminance bounds, that is, the two extreme states (black and white), the percept of the illusion is not actually oscillating between equal perceptual bounds relative to the neutral gray (i.e., mean display luminance). In other words, the balanced luminance extremes imposed by the nulling procedure may not correspond to the extreme lightness induction states a person actually perceives, which makes it difficult to identify and report a steady nulling level over time. Recently, evidence from multiple studies have now shown that striate cells in non-human primates actually respond more strongly to uniformly black stimuli than white, with the difference being more prevalent at the stimulus center than at the edges (*Kremkow et al., 2014*; *Xing, Yeh & Shapley, 2010*; *Zurawel et al., 2014*). Therefore, the assumption that both extreme static states of a lightness illusion induce an equal lightness percept change (compared against a context-free neutral gray) may not be true and requires further investigation. Any evidence for an asymmetry in lightness induction toward lighter or darker percepts, despite the fact that the gray of the physical target is exactly in the middle of the maxima and minima, could reveal other perceptual biases of the human visual system.

## CONCLUSIONS

This study explicitly examined the existence of a critical change in lightness induction between static and continuous modulation states, while also exploring the extent to which different perceptual grouping principles and figure-ground cues alter the critical change of lightness induction in general within a low frequency modulation regime. Significant increases in lightness induction were found when comparing static against continuous surround modulation conditions for most stimulus types. Further increasing the continuous surround modulation frequency was also found to significantly decrease the

lightness induction effect for certain stimulus types, while remaining unaltered for other stimulus types.

## Study limitations

The fixed stimulus size and spatial frequency content in this study present a potential limitation. It is unclear if the results reported here under continuous modulation viewing conditions are resilient across these dimensions, or highly contingent upon them (*Rossi & Paradiso, 1996*; *Salmela & Laurinen, 2009*; *Shi et al., 2013*). Similarly, the luminance range over which the stimuli were manipulated is low and small relative to natural viewing conditions and scene statistics (*Frazor & Geisler, 2006*; *Radonjić et al., 2011*). Whether or not low frequency lightness modulation persists across a wide range of absolute luminance spectrums remains to be investigated.

## Future directions

The nulling paradigm employed in this study is easily transferable to a neuroimaging environment, whereby participants could be presented nulled (lightness induction absent) and non-nulled (lightness induction present) stimulus conditions. Population neural activity across early visuocortical areas could then be compared across continuous surround modulation frequency conditions to make inferences about the dependency of hierarchical cortical recruitment on the lightness perception and the strength of figure-ground cues and modulation frequency. Furthermore, it would be possible to differentiate sub-cortical and cortical areas which respond to either physical luminance or lightness induction modulation, or both, providing a better understanding of the emergence of lightness perception in humans. Lastly, there exist more geometrically complex lightness induction illusions, for example the Craik-O'Brien-Cornsweet illusion (*Davey, Maddess & Srinivasan, 1998*) and Benary's cross (*Benary, 1924*), which could be further studied under temporal modulation conditions. The large variety of feature induction illusions, including color and contrast induction (*Zaidi, Spehar & Shy, 1997*), which remain to be investigated under static and continuous modulation presentation, could also offer further insight into how humans perceive more complex objects and scenes.

# ACKNOWLEDGEMENTS

We would like to expressly thank Dr. Shelley Russek, Sandra Grasso, Dr. Michael Lyons, and the members of the Computational Neuroscience & Vision lab for all their help and support. The authors of this manuscript declare no conflicts of interest.

## Funding

The authors received no funding for this work.

## Competing Interests

The authors declare that they have no competing interests.

## Author Contributions

- Louis Nicholas Vinke conceived and designed the experiments, performed the experiments, analyzed the data, prepared figures and/or tables, authored or reviewed drafts of the paper, and approved the final draft.
- Arash Yazdanbakhsh conceived and designed the experiments, authored or reviewed drafts of the paper, suggested the previous studies to be reviewed and included, and approved the final draft.

## Human Ethics

The following information was supplied relating to ethical approvals (i.e., approving body and any reference numbers):

Boston University Charles River Campus Institutional Review Board approved this study (3651E).

## Data Availability

The raw dataset is available as a Supplemental File.

## Supplemental Information

Supplemental information for this article can be found online at http://dx.doi.org/10.7717/peerj.8918#supplemental-information.

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
