# Peer review of "Lightness induction enhancements and limitations at low frequency modulations across a variety of stimulus contexts"

_PeerJ, doi:10.7717/peerj.8918_

## Round 0.1 · original submission · Minor Revisions

Your paper has now been seen by two reviewers who seem quite pleased with the study as a whole but who both suggest minor revisions. Please now take the time to address these concerns fully.

Reviewer 1 ·

Basic reporting

The writing is, for the most part, clear and unambiguous, with the following exceptions:
1. The main "nulling" paradigm is not well defined. Referring to line 136-137: "Specifically, participants were asked to null any perceived modulation in lightness" - it is best not to define nulling with the term itself, but substitute it with other words, such as cancel out, or extinguish.
2. Do the authors mean to use “static” rather than discrete? Line 138: “the surrounding lightness illusion stimulus was modulated discretely or continuously in time”. Discretely here would imply step-wise increments of changes to lightness, which I believe is not what the authors did. They would better describe their method using “static” instead of “discrete”. This applies to all uses of “DSM” and “discrete static modulation”. “Modulation” implies change to the stimulus, which was not performed. This looks like a static condition (described as 0 Hz).

In regards to referencing, the nulling paradigm is not properly cited. It is referred to, and defined, in Adelson (1993) but not in Blakeslee & McCourt (2008). Nulling is also used in Adelson (1993) with a static image, so the nulling paradigm here needs to either refer to another reference with temporal modulation, or clarify the difference between previously published methodologies and what is used here.

Figures are in general clear. Error bars are used and defined. Figure 1B: are there supposed to be c annotations on the white checkerboard squares? Otherwise the luminance inversion is not consistent in this figure.

Experimental design

There would be more consistency if all of the stimuli were the same size, because then you would also be controlling for the size of the surround area. Does the size of the effect for Munker-White change when presented using a square aspect ratio?

For the Simultaneous Contrast (SC) effect, an area 8 times the target patch is modulated, which is the same for the checkerboard, so it appears that these two stimuli are more directly comparable in terms of changes to luminance given fixed area. With Munker-White (MW), if the “flankers” are modulated, this is approximately 4 times the target patch area, which also differs if the “assimilators” are modulated, which are approximately 10 times the patch area. If the size of the effect is dependent upon the surround area, then the flanker effect would be weaker than the assimilator effect, which is shown here. Please justify your reasoning for choosing a different size stimuli for MW.

Validity of the findings

The authors compare static to continuously modulated stimuli, but do not clearly state the differences in the rank order of the effect between these two conditions. In Figure 2, in the static condition, the strongest effect is shown by SC, then checkerboard, then MW. This rank-order changes with the temporal modulation, such that SC still remains with having the highest effect, but is now followed by MW and then checkerboard. This suggests different mechanisms occurring between the static versus continuously modulated stimuli. The figure-ground effect would remain the same between static versus continuously modulated stimuli. What other mechanisms would you propose could cause this difference?

Reviewer 2 ·

Basic reporting

1. The authors do not cite any literature investigating the role of figure-ground on lightness induction, nor do they explain the figure-ground theories in depth, despite stating several times that there are numerous sources and strong support for the theories. It would greatly strengthen their argument if they included this. Additionally, it seems that most literature investigating the role of figure-ground cues in lightness induction talk about perceptual grouping as a whole, not specifically figure-ground (Gilchrist, 1999). It would be more fitting to use something like ‘perceptual grouping’ instead of ‘figure-ground’ in the text.
2. Figure 1C is a little bit confusing, given that the authors include the luminance inversion symbol when in reality, those are not the two states it is fluctuating between.
3. Typos/grammar issues:
Throughout: should be a space before the first parenthesis when citing.
Throughout: Some phrasing is a bit repetitive.
Lines 36-39: Awkward, repetitive phrasing.
Line 68: ‘modulations’ should be ‘modulation’.
Line 68: There should not be a comma after ‘studies’.
Line 71: There should not be a comma after ‘states’.
Line 74: There should not be a comma after ‘understood’.
Line 81: ‘T- junctions’ should be ‘T-junctions’.
Line 106: Add ‘the’ before ‘lightness’.
Line 190-191: Parentheses are in the incorrect spots.
Lines 188-191: Awkward, repetitive phrasing. Delete last phrase.
Line 205: There should not be a comma after ‘lightness’.
Line 205: Delete ‘by’.
Line 208: Delete ‘, which is’. There should also be no comma there.
Line 220: ‘Michselson’ should be ‘Michelson’.
Line 240: There should not be a comma before ‘strongly’.
Line 254: ‘found’ should be ‘find’.
Line 258: There should be a comma before ‘we used’.
Line 314: There should be a comma after ‘presentation’.

Experimental design

1. The authors use discrete, manual switching (DSM) between extremes of an illusion and call it the ‘0 Hz’ condition (in comparison to the continuous modulation (CSM) conditions). While the inclusion of these trials is useful, there are substantial significant differences between the DSM and CSM conditions. Given that observers are required to fixate at a static illusion for an extended period of time, adaption is likely to play a huge role in the perception of brightness. The authors ought to bring up this caveat and consistently distinguish these trials from the CSM trials. It is a bit of a stretch to label them as the ‘0 Hz’ conditions and for their conclusions to depend strongly on this assumption.
2. Figure 2: The statistics here seem awkward. Looking at differences between slopes in Fig 2A, it would be difficult to believe that MWf and SC have slopes significantly different from 0, yet MWa does not. Or, more specifically, do they really think that there is a difference between MWf and MWa?
3. The authors should indicate how well the linear functions fitted the data and plot them, perhaps in Fig 2A. Considering that it neither hurts nor helps their claim of figure-ground involvement, the statement that MWa does not have a decreasing trend while MWf and SC do is misleading.
4. Lines 214: Readers would have a better idea of the authors’ predictions if they plotted the potential ‘inverted-U’ shape from the JNTM.
5. Lines 226-232: It is probably best to use effect sizes to describe the magnitude of the difference between the DSM and slowest CSM nulling amplitudes, instead of just comparing mean amplitudes.

Validity of the findings

1. The authors are convinced that the degree of figure-ground segmentation in particular influences lightness induction, based on their comparison of three configurations with differing figure-ground ambiguity. However, these three illusions differ in more ways that just figure-ground ambiguity. Particularly, they have power at different spatial frequencies. There are numerous spatial filtering theories that have been able to explain most brightness induction effects (e.g., Blakeslee and McCourt, Dakin and Bex), yet the authors do not mention them, let alone challenge them.

Additional comments

The subject of temporal modulation influences on lightness induction is an interesting and worthy topic, and the authors point out a significant hole in the literature.

---

## Round 0.2 · accepted · Accept

Thanks for your submission and revisions! It's a fine paper. Congratulations!